# Containerized Distributed Value-Based Multi-Agent Reinforcement Learning

## Abstract

Multi-agent reinforcement learning tasks put a high demand on the volume of training samples. Different from its single-agent counterpart, distributed value-based multi-agent reinforcement learning faces the unique challenges of demanding data transfer, inter-process communication management, and high requirement of exploration. We propose a containerized learning framework to solve these problems. We pack several environment instances, a local learner and buffer, and a carefully designed multi-queue manager which avoids blocking into a container. Local policies of each container are encouraged to be as diverse as possible, and only trajectories with highest priority are sent to a global learner. In this way, we achieve a scalable, time-efficient, and diverse distributed MARL learning framework with high system throughput. To own knowledge, our method is the first to solve the challenging Google Research Football full game 5_v_5. On the StarCraft II micromanagement benchmark, our method gets $4-18\times$ better results compared to state-of-the-art non-distributed MARL algorithms.

## 1 Introduction

Deep reinforcement learning (DRL) has been proved effective in various complex domains, including the game of Go (Silver et al., 2017), Dota (OpenAI, 2018), and StarCraft II (Vinyals et al., 2017). However, as problems increase in scale, the training of DRL models is increasingly time-consuming (Mnih et al., 2016), which is partly because complex tasks typically require large amounts of training samples. Distributed learning provides a promising direction to significantly reduce training costs by allowing the RL learner to effectively leverage massive experience collected by a large number of actors interacting with different environment instances.

Compared to its single-agent counterparts, multi-agent tasks involve more learning agents and a larger search space consists of joint action-observation history. Therefore, multi-agent reinforcement learning (MARL) typically requires more samples (Kurach et al., 2020) and is more time-consuming. However, most of the impressive progress achieved by distributed DRL focuses on the single-agent setting and leaves the efficient training of MARL algorithms largely unstudied. To makeup for this shortage, in this paper, we study distributed value-based multi-agent reinforcement learning. We first pinpoint three unique challenges posed by the increasing number of agents in multi-agent settings listed as follows.

(1) Demanding data transfer. Coming with the increased number of agents is a larger volume of experience – modern MARL algorithms typically require local observation history of all agents apart from global states. If we deploy environments on CPUs like state-of-the-art single-agent distributed DRL algorithms (Espeholt et al., 2019), the data transfer between CPUs and GPUs will consume significant CPU computation utility. (2) Inter-process communication. Inter-process communication between environments, buffers, and learners is more likely to block in multi-agent settings. This is because the experience volume is large and reading/rewriting becomes more time-consuming. (3) Exploration. The search space of multi-agent problems grows exponentially with the number of agents. Learning performance of MARL algorithms largely rely on efficient exploration in such a large space.

In this paper, we propose a containerized distributed value-based multi-agent reinforcement learning framework (CMARL) to address these problems. We pack several actors interacting with environments, a local learner, a local buffer, and a carefully designed multi-queue manager into a *container*.

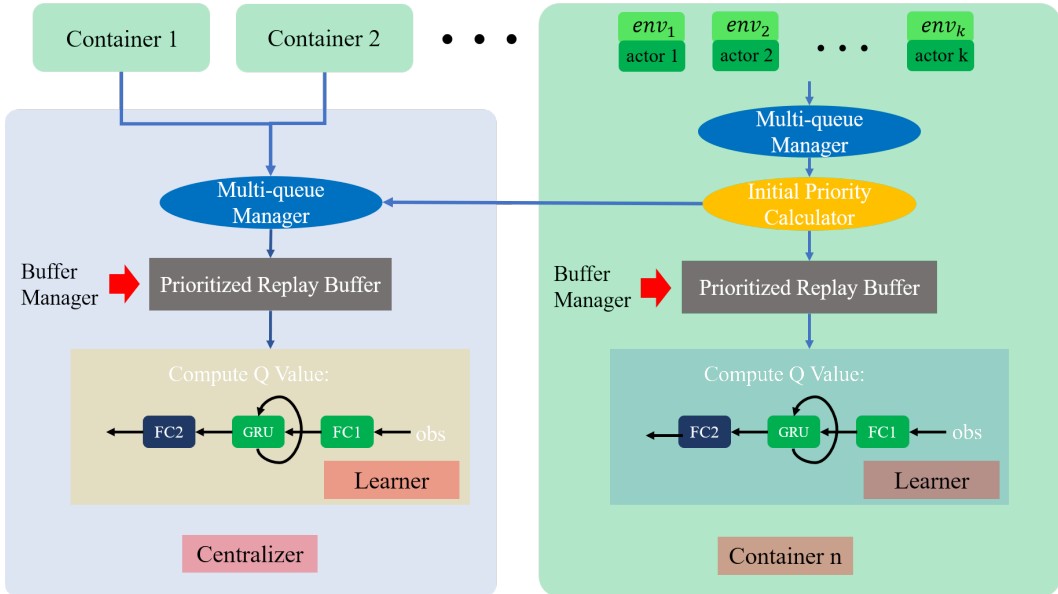

Figure 1: Containerized value-based multi-agent reinforcement learning architecture.

A container interacts with its environment instances, collects data, avoids block via its multi-queue manager, and actively updates its local policy. Data with high priority is transferred to a global learner that brings together the most talented and valuable experience to train the target policy. We further share the parameters of shallow layers for local and global learners to accelerate training while encouraging deep layers of local learners to be as diverse as possible to improve exploration.

The advantages of the CMARL framework are as follows. (1) A container can be deployed on either CPUs or a GPU, making our method scalable and adaptive with the available computational resources. (2) Largely reduced data transfer. Data collection and prioritization happens within the container, and only highly prioritized data is transferred. (3) Multi-queue managers work asynchronously from policy learning, enabling efficient and unblocked inter-process communication when data collection. (4) Diverse behavior enables efficient exploration in the large search space.

In this way, CMARL makes a *scalable*, *time-efficient*, and *diverse* distributed MARL framework with *high system throughput*. We empirically evaluate the performance of our architecture on both Google Research Football (Kurach et al., 2020) and StarCraft II micromanagement challenges (Samvelyan et al., 2019). Our method is the only algorithm that can obtain a positive goal difference on the three GRF tasks, and is the first algorithm which can solve the challenging 5_v_5 full game. On the SMAC benchmark, our method obtains $4-18\times$ better results compared against state-of-the-art non-distributed MARL algorithms.

## 2 METHODS

In this section, we present our novel containerized framework (Fig. 1) for distributed MARL. We consider fully cooperative multi-agent tasks that can be modelled as a Dec-POMDP (Oliehoek et al., 2016) consisting of a tuple $G=\langle I, S, A, P, R, \Omega, O, n, \gamma \rangle$, where $I$ is the finite set of $n$ agents, $\gamma \in [0, 1)$ is the discount factor, and $s \in S$ is the true state of the environment. At each timestep, each agent $i$ receives an observation $o_i \in \Omega$ drawn according to the observation function $O(s, i)$ and selects an action $a_i \in A$, forming a joint action $\boldsymbol{a} \in A^n$, leading to a next state $s'$ according to the transition function $p(s'|s, \boldsymbol{a})$, and observing a reward $r = R(s, \boldsymbol{a})$ shared by all agents. Each agent has local action-observation history $\tau_i \in \mathrm{T} \equiv (\Omega \times A)^*$.

Distributed deep reinforcement learning aims to provide a scalable and time-efficient computational framework. In the multi-agent setting, distributed RL encounters unique challenges.

(1) The experience of MARL agents typically consists of local observations, and actions of all agents. Moreover, the centralized training with decentralized execution paradigm (Foerster et al., 2017; Rashid et al., 2018) requires global states that contain information of all agents. Therefore, the size of experience grows quadratically with the number of agents. Consequently, transferring experience across different devices is very consuming, leading to a very high CPU usage. Meanwhile, large volume experience may also cause block in inter-process communication. In order to reduce such overhead, we pack several actors and environments into one container and design a multi-queue manager to avoid blocking. The container only sends a portion of experience to the centralized learner, and the centralized learner learns a global policy from these experiences. The whole architecture consists of several containers and one centralized learner.

(2) Another challenge of multi-agent reinforcement learning is that the action-observation space grows exponentially with the number of agents, posing a great challenge to exploration. Multiple containers allow us to do diverse exploration in a way that each container learns an individual policy different from others and uses that policy to explore.

Therefore, the proposed containerized framework (Fig. 1) holds the promise to solve these unique challenges of distributed multi-agent reinforcement learning. However, to realize this goal, many structural details need to be designed for each framework component, including the container, the centralized learner, and the training scheme. We now describe them in greater detail.

## 2.1 CONTAINER

Inside each container, there are $k$ actors interacting with $k$ environment instances to collect experience, one container buffer storing experience, and one container learner training the container policy with batches of trajectories sampled from the local buffer.

A critical design consideration of a successful distributed reinforcement learning framework is the uninterrupted learning of learners. To this end, we need to constantly sample batches from the container buffer. Meanwhile, the buffer needs to update itself with new experience constantly. In order to well manage these two operations and avoid read/write conflicts, we introduce a buffer manager, which is the only process controlling the buffer.

The buffer manager repeatedly inserts new experience and samples batches. Directly sending new experience from actors to the buffer manager has two shortcomings. (1) The new experience would be stacked in the multi-process data-transfer queue when the buffer manager is sampling. In this case, actors have to wait, without collecting new trajectories until the buffer finishes sampling. Consequently, experience collection is slowed down. (2) Receiving trajectories one by one is slower than receiving trajectories in a batch. Moreover, receiving and inserting new experience frequently increases the waiting time between batch sampling. As sampling has to wait, the container learner has to wait when there is no batch to train. These two shortcomings become particularly problematic in multi-agent settings as trajectories consume more space in the multi-process data-transfer queue.

In order to tackle these issues, we introduce a multi-queue manager. There is a shared signal between the multi-queue and the buffer manager. The multi-queue manager constantly gathers new experience together unless the signal indicates that the buffer manager requires new batches. When the signal is set, the multi-queue manager compacts all new experience it has gathered to a batch and send them to the buffer manager. Since we use prioritized experience replay, this batch goes through an initial priority calculator before the buffer manager where its priority is calculated.

The priority of one trajectory is computed by $p_\tau = \text{Normalize}(\sum_t r_t) + \epsilon$, where $\text{Normalize}(X) = \frac{X-L}{H-L}$, and $L, H$ are the lower and upper bound of the sum of rewards in a whole trajectory, respectively. $\epsilon$ is a small constant that avoids a zero probability during sampling.

## 2.2 CENTRALIZER

The centralizer has an architecture similar to the container (Fig. 1), except that new experience comes from containers rather than actors. After containers' initial priority calculator receive a batch of new experience and compute their priority, containers will send $\eta\%$ of the experience to the centralizer's experience receiver. Transferred experience is sampled with a probability proportional to the priority.

Here, $\eta$ is a real number between 0 and 100, indicating the fraction of experience to be sent to the centralized learner.

Although our architecture can be combined with any value-based MARL algorithms, in this paper, we use QMIX (Rashid et al., 2018) as the underlying algorithm. The centralized learner updates a QMIX network. Specifically, agents share a three-layer local Q-network, with a GRU (Cho et al., 2014) between two fully-connected layers, and the global Q value $Q_\theta(\boldsymbol{\tau}, \boldsymbol{a})$, parameterized by $\theta$, is learned as a monotonic combination of local Q values. The centralized learner is updated by the following $TD$ loss:

$$\mathcal{L}_{TD}(\theta) = \mathbb{E}_{\mathcal{B} \sim \mathcal{D}_{\text{cen}}} \left[ \frac{\sum_{\boldsymbol{\tau} \in \mathcal{B}} \sum_{t=0}^{T_{\boldsymbol{\tau}}-1} \left[ Q_\theta(\boldsymbol{\tau}_t, \boldsymbol{a}_t) - (r_t + \gamma \max_{\boldsymbol{a}} Q_{\theta'}(\boldsymbol{s}_{t+1}, \boldsymbol{a})) \right]^2}{\sum_{\boldsymbol{\tau} \in \mathcal{B}} T_{\boldsymbol{\tau}}} \right] \tag{1}$$

where $T_{\boldsymbol{\tau}}$ is the length of trajectory $\boldsymbol{\tau}$, $\mathcal{D}_{\text{cen}}$ is the centralized buffer, the expectation means that batches are sampled from $\mathcal{D}_{\text{cen}}$ according to the priority of trajectories, and $\theta'$ is parameters of a target network copied from $\theta$ every C Q-network updates.

## 2.3 ENCOURAGING DIVERSITY AMONG CONTAINERS

Using multiple containers to interact with the environment leads to more time-efficient experience collection. Training will be boosted in large exploration spaces posed by multi-agent tasks when the collected experience is diverse. Such diversity can be achieved by letting containers act and explore differently from each other.

In our architecture, containers maintain a Q-network with the same architecture as the centralized learner. Although the architecture is the same, container Q-networks are learned individually and encouraged to be different. To explicitly encourage diversity among containers, in addition to the local TD loss, every container includes a diversity objective in the loss function to maximize the mutual information between container id and its local experience

$$I(\boldsymbol{\tau}, id) = \mathbb{E}_{\boldsymbol{\tau}, id}[\log \frac{p(\boldsymbol{\tau}|id)}{p(\boldsymbol{\tau})}]. \tag{2}$$

We expand the mutual information as follows:

$$I(\boldsymbol{\tau}, id) = \mathbb{E}_{\boldsymbol{\tau}, id} \left[ \log \frac{p(\boldsymbol{o}_0|id)}{p(\boldsymbol{o}_0)} + \sum_t \log \frac{p(\boldsymbol{a}_t|\boldsymbol{\tau}_t, id)}{p(\boldsymbol{a}_t|\boldsymbol{\tau}_t)} + \sum_t \log \frac{p(\boldsymbol{o}_{t+1}|\boldsymbol{\tau}_t, \boldsymbol{a}_t, id)}{p(\boldsymbol{o}_{t+1}|\boldsymbol{\tau}_t, \boldsymbol{a}_t)} \right] \tag{3}$$

Here, $\boldsymbol{o}_0$ is the initial observation, and its distribution is independent of container id. Therefore, $p(\boldsymbol{o}_0|id) = p(\boldsymbol{o}_0)$. Similarly, $p(\boldsymbol{o}_{t+1}|\boldsymbol{\tau}_t, \boldsymbol{a}_t)$ is decided by the transition function and is the same for all containers. So we have $p(\boldsymbol{o}_{t+1}|\boldsymbol{\tau}_t, \boldsymbol{a}_t, id) = p(\boldsymbol{o}_{t+1}|\boldsymbol{\tau}_t, \boldsymbol{a}_t)$. It follows that

$$I(\boldsymbol{\tau}, id) = \mathbb{E}_{\boldsymbol{\tau}, id} \left[ \sum_t \log \frac{p(\boldsymbol{a}_t|\boldsymbol{\tau}_t, id)}{p(\boldsymbol{a}_t|\boldsymbol{\tau}_t)} \right]. \tag{4}$$

However, $p(\boldsymbol{a}_t|\boldsymbol{\tau}_t, id)$ is typically a distribution induced by $\epsilon$-greedy, distinguishing only the action with the highest probability and concealing most information about value functions. Therefore, we use the Boltzmann SoftMax distribution of local Q values to replace $p(\boldsymbol{a}_t|\boldsymbol{\tau}_t, id)$ and optimize a lower bound of Eq. 4:

$$I(\boldsymbol{\tau}, id) \geq \mathbb{E}_{\boldsymbol{\tau}, id} \left[ \sum_t \log \frac{\boldsymbol{\pi}_{id}(\boldsymbol{a}_t|\boldsymbol{\tau}_t)}{p(\boldsymbol{a}_t|\boldsymbol{\tau}_t)} \right], \tag{5}$$

where $\boldsymbol{\pi}_{id}(\boldsymbol{a}_t|\boldsymbol{\tau}_t) = \Pi_{i=1}^n \pi_{id}^i(a_t^i|\tau_t^i)$. The inequality holds because $D_{\text{KL}}[\boldsymbol{\pi}_{id}(\boldsymbol{a}_t|\boldsymbol{\tau}_t) \| p(\boldsymbol{a}_t|\boldsymbol{\tau}_t, id)]$ is non-negative. We approximate $p(\boldsymbol{a}_t|\boldsymbol{\tau}_t) = \Pi_{i=1}^n p^i(a_t^i|\tau_t^i)$ by $p(\boldsymbol{a}_t|\boldsymbol{\tau}_t) \approx \Pi_{i=1}^n (\frac{1}{N} \sum_{j=1}^N \pi_j(a_t^i|\tau_t^i))$, where $N$ is the number of containers. Then we get the lower bound of Eq. 4 to optimize:

$$I(\boldsymbol{\tau}, id) \geq \mathbb{E}_{\boldsymbol{\tau}, id} \left[ \sum_t \sum_{i=1}^n \log \frac{\pi_{id}(a_t^i|\tau_t^i)}{\frac{1}{N} \sum_{j=1}^N \pi_j(a_t^i|\tau_t^i)} \right] \tag{6}$$

$$= \mathbb{E}_{\boldsymbol{\tau},id} \left[ \sum_t \sum_{i=1}^n D_{\mathrm{KL}} \left[ \pi_{id}(\cdot|\tau_t^i) \| \frac{1}{N} \sum_{j=1}^N \pi_j(\cdot|\tau_t^i) \right] \right] \tag{7}$$

In practice, we minimize the following loss for training the learner of the $id$-th container:

$$\mathcal{L}(\theta_i; \mathcal{B}) = \mathcal{L}_{TD}(\theta_i) + \beta \left[ \frac{1}{|\mathcal{B}|} \sum_{\tau \in \mathcal{B}} \sum_t \sum_{i=1}^n D_{\mathrm{KL}} \left[ \pi_{id}(\cdot|\tau_t^i) \| \frac{1}{N} \sum_{j=1}^N \pi_j(\cdot|\tau_t^i) \right] - \lambda \right]^2 \tag{8}$$

where $\theta_i$ is the parameters of the container Q-function, $\beta$ is a scaling factor, and $\lambda$ is a factor controlling the value of the KL divergence.

To balance diversity and learning sharing, we split agent network in containers into two parts. The lower two layers are shared among the global learner and all containers. Training data is sampled from the global buffer, and weights are periodically copied to the containers every $t_{\mathrm{global\_update\_time}}$ seconds to avoid frequent transfer and unstable learning in containers. The last layer differs in each container and is updated locally, which enables containers to act differently under the same observation.

## 3 RELATED WORKS

Distributed single-agent RL algorithms have been a popular research area in recent years, among which the search for distributed value-based methods has been quite fruitful. The first pioneering value-based architecture, GORILA (Nair et al., 2015), adapts the Deep Q-Network (DQN) for distributing acceleration. It introduces several innovative ideas, such as isolated actors and learners running in parallel, distributed neural networks, and distributed replay buffer storage, but is not widely used for poor time efficiency and sample efficiency caused by asynchronous SGD. Later, APE-X (Horgan et al., 2018) accelerates training by separating data collection from learning, allowing massive scales of samples to be generated and prioritized in a distributed manner. R2D2 (Kapturowski et al., 2018) studies distributed training of recurrent policies to fit partially observable environments better. SEED (Espeholt et al., 2019) features a centralized inference interface within the learner module to reduce the cost of transmitting network weights frequently.

Another popular approach towards distributed single-agent RL is to adapt policy gradient methods to paralleled training. A3C (Mnih et al., 2016) first introduces a general asynchronous training architecture for actor-critic methods, in which workers retrieve network weights from a parameter server, independently interact with the environment to calculate gradients, and then send the gradients back to the parameter server for centralized policy updates. DPPO (Heess et al., 2017) is the distributed implementation of PPO using a similar method of distributed gradient gathering. On the other hand, IMPALA (Espeholt et al., 2018) introduces an off-policy correction method (called the V-trace method), where the actor-learner architecture is utilized, and experience trajectories are communicated instead of gradients. Compared to A3C, it displays desirable properties of stabler convergence, better scalability and higher robustness.

Reinforcement learning in multi-agent settings is becoming increasingly important for complex real-world decision-making problems, and has been seeing significant progresses recently (Lowe et al., 2017; Rashid et al., 2018; Wang et al., 2020; 2021b). Li et al. (2021) encourages diversity for efficient multi-agent reinforcement learning and thus is related to our work. The difference lies in that we encourage diverse behavior among containers, instead of among agents as in (Li et al., 2021). It is evident that the training of multi-agent RL generally takes more time and computational resources than that of single-agent RL, and thus distributing acceleration is more urgently demanded. Nevertheless, the demand is largely ignored, as little existing literature has looked into the design of general distributed MARL architectures. In this paper, we develop a novel architecture to address this challenge, which significantly improves the scalability of value-based MARL algorithms on distributed systems.

# 4 EXPERIMENTS

Our experiment is aimed at answering the following problems: (1) Can our containerized distributed framework accelerate the training of MARL agents? How much improvement can it bring compared to non-distributed state-of-the-art algorithms? (2) How does the time efficiency of training improves when given more computational devices?

**Baselines.** We compare our method with various baselines shown in Table 1. The baselines include state-of-the-art multi-agent value decomposition methods (QMIX, Rashid et al. (2018), QPLEX, Wang et al. (2021a)), identity-aware diversity-encouragement method (CDS, Li et al. (2021)), and role-based method (RODE, Wang et al. (2021b)). We show the median and variance of the performance for our method and baselines tested with five random seeds.

Table 1: Baseline algorithms and ablations.

|  | Alg. | Description |
|---|---|---|
| Related Works | QMIX | Rashid et al. (2018) |
|  | QPLEX | Wang et al. (2021a) |
|  | RODE | Wang et al. (2021b) |
|  | CDS | Li et al. (2021) |
| Ablations | CMARL_no_diversity | actors explore with the centralised policy |
|  | CMARL_2_containers | 2 containers, each 13 actors |
|  | CMARL_1_container | 1 container, 13 actors |
|  | CMARL_8_actors | 3 containers, each 8 actors |
|  | CMARL_2_actors | 3 containers, each 2 actors |
| Other Distributed Medhos | QMIX-BETA | Parallel implementation of QMIX, 39 actors. |
|  | APEX | APE-X (Horgan et al., 2018) architecture, 10 actors |
|  | APEX-overload | APE-X (Horgan et al., 2018) architecture, 14 actors for `corridor`, 16 actors for `6h_vs_8z`, 30 actors for GRF. |

**Experiment scheme.** We fine-tune hyperparameters on `corridor` from the StarCraft II micromanagement (SMAC) benchmark (Samvelyan et al., 2019) and `academy_counterattack_easy` from the GRF environment. Then we use the same hyperparameters as `corridor` to run experiments on other super hard SMAC scenarios and the same hyperparameters as `academy_counterattack_easy` to run experiments on other GRF environments. We did not tune hyperparameters for experiments on easy and hard maps of the SMAC benchmark.

## 4.1 OVERALL PERFORMANCE ON GOOGLE RESEARCH FOOTBALL AND STARCRAFT II

We benchmark our method on 3 Google Research Football (GRF) scenarios (`5_vs_5`, `academy_counterattack_easy`, and `academy_counterattack_hard`) and 11 scenarios from the SMAC benchmarks (four super hard scenarios: `3s5z_vs_3s6z`, `6h_vs_8z`, `MMM2`, `corridor`, two hard scenarios: `5m_vs_6m`, `3s_vs_5z`, and five easy scenarios: `2s_vs_1sc`, `2s_vs_3z`, `3s_vs_5z`, `1c3s5z`, `10m_vs_11m`). To have an overview of our performance, we average performance on different scenarios for both GRF and SMAC. Because different scenarios needs different training time, we align learning curves by computing the ratio of the elapsed training time to the total training time for each scenario and then take the average. The results are shown in Fig. 2.

For scenarios of GRF, we plot the average goal difference, which is the number of goals scored by MARL agents minus the number of goals scored by the other team. The overall performance is shown in Fig. 2 left. After training for more than half of the total training time, our method starts to significantly outperforms baseline algorithms.

The overall performance of SMAC is shown in Fig. 2 right. The overall winning rate of our method is at least four times better than any baseline algorithm ($4\times$ against QPLEX, $4.5\times$ against QMIX, $10\times$ against RODE, and $18\times$ against CDS). We also notice that RODE and CDS perform better on super hard scenarios, but not on easy maps, which lead to the overall unsatisfactory results.

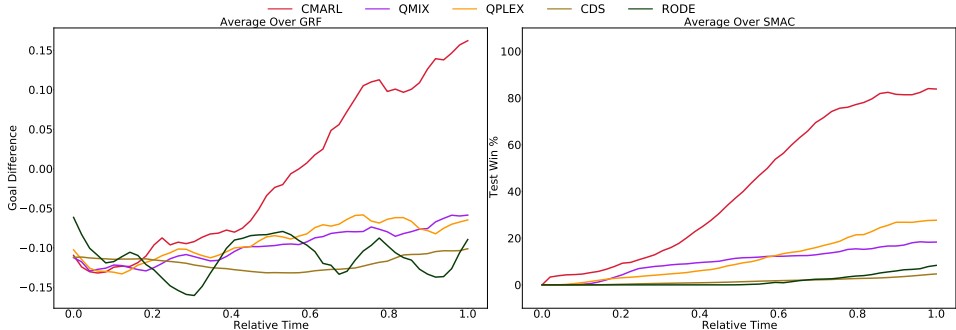

Figure 2: Left: Average goal difference of our approach and baseline algorithms on 3 Google Research Football scenarios. Right: Average winning rate of our approach and baseline algorithms on 11 SMAC scenarios. The x-axis is the ratio of the elapsed training time to the total training time.

## 4.2 PERFORMANCE ON GOOGLE RESEARCH FOOTBALL

In this section, we test our method on three challenging Google Research Football (GRF) scenarios: `academy_counterattack_easy`, `academy_counterattack_hard`, and `5_vs_5`. In the first two `academy counterattack` scenarios, four agents start from the opposition's half and attempt to score against one (`easy`) or two (`hard`) defenders and an opposing goalkeeper. In the `5_vs_5` scenario, agents play a 15-minute regular game, including offense, defense, out-of-bounds, and offside. This task has a very large search space. In GRF scenarios, the purpose of all agents' actions is to coordinate timing and positions sophisticatedly to seize valuable opportunities of scoring.

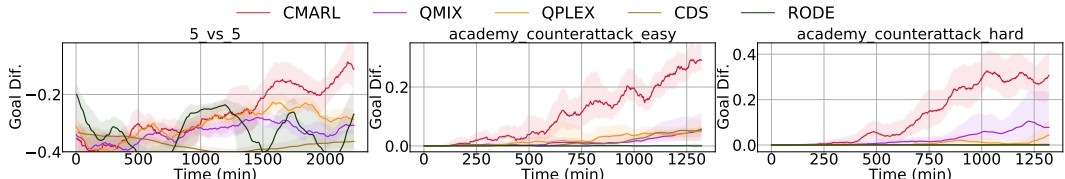

Figure 3: Comparison of our approach against baseline algorithms on Google Research Football.

We show the performance comparison against baselines in Fig. 3. Our method outperforms in all the scenarios with a stable training process. In the `academy_counterattack_easy` scenario and `academy_counterattack_hard` scenario, the game terminates after our agents scoring or losing the ball, so the goal difference can only be 0 or 1. RODE and CDS overcome the negative effects of QMIX or QPLEX's monotonicity with roles and identity-aware diversity, respectively. However, these methods need significantly more time to demonstrate their advantage. With the benefits of our containerized distributed framework, our method can sample considerably diverse trajectories. Our container priority calculator can further choose critical trajectories for training and sharing, significantly improving learning efficiency. In the `5_vs_5` scenario, the game ends only when the time step reaches the limit. As demonstrated in Fig. 3 left, baselines struggle to learn sophisticated policies and suffer from performance degradation. Our method can stably scale to highly complex policies with the advantages of our containerized distributed framework.

## 4.3 PERFORMANCE ON STARCRAFT II

In this section, we test our approach on the StarCraft II micromanagement (SMAC) benchmark (Samvelyan et al., 2019). In this benchmark, maps have been classified as easy, hard and super hard. As shown in Fig. 4, we present the results for four super hard scenarios: `3s5z_vs_3s6z`, `6h_vs_8z`, `MMM2`, `corridor`, two hard scenarios: `5m_vs_6m`, `3s_vs_5z`, and five easy scenarios: `2s_vs_1sc`, `2s_vs_3z`, `3s_vs_5z`, `1c3s5z`, `10m_vs_11m`. For the four super hard scenarios, our approach outperforms all baselines in learning efficiency. Especially for `6h_vs_8z` and `MMM2`, our method achieves remarkable performance in less then four hours, while baselines, such as RODE and

CDS, achieve comparable performance in more than one day. This significant reduction in learning time indicates the great advantages of our method for practical application, which sheds light on introducing MARL algorithms to real world and for solving challenging problems. For other hard and easy scenarios, our method maintains its advantages of time efficiency and training stability, indicating our method's wide applicable range.

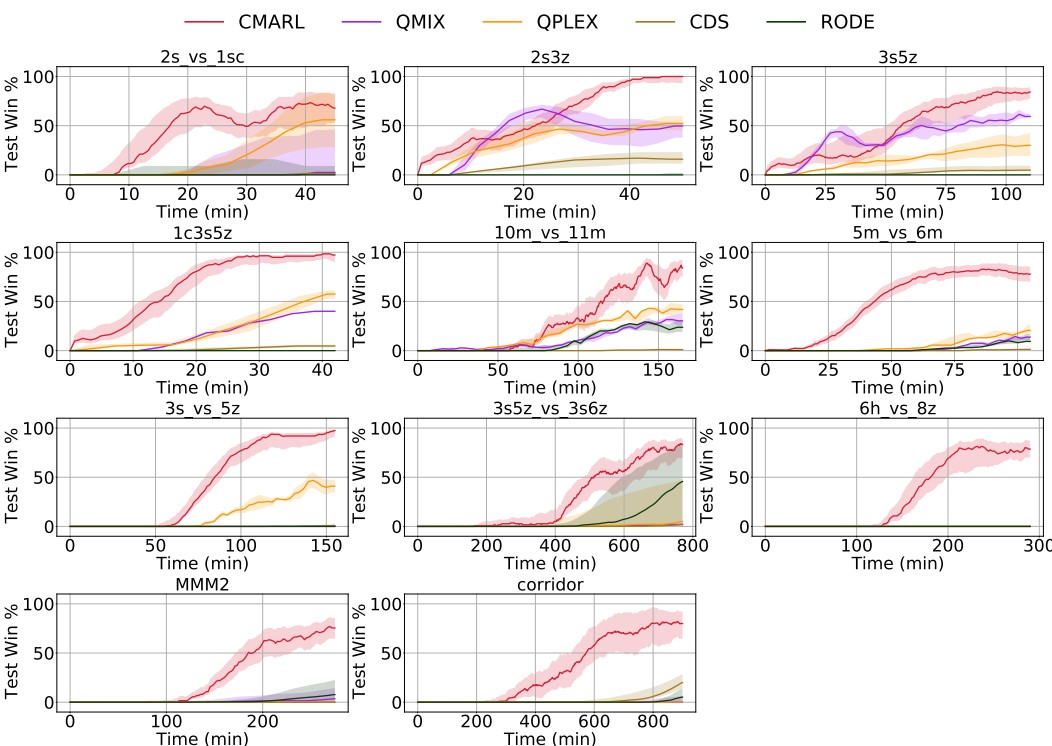

Figure 4: Comparison of our approach against baseline algorithms on SMAC scenarios.

## 4.4 ABLATION STUDIES

Our framework mainly contributes two novelties: (1) The container architecture facilitating efficient data collection, transfer, and training of value functions. (2) Encouraging diversity of experience collected from different containers. In this section, we carry out ablation studies to show the contribution of these novelties to the final performance improvement provided by our method, which can explain why our method can efficiently explore and utilize computation resources. Ablation studies are carried out on one super map from the SMAC benchmark `corridor` and a map from the GRF environment `academy_counterattack_hard`.

For (1), to test whether the container architecture can help utilize computation resources well, we conduct experiments by running our framework with different numbers of total actors (the sum of the number of actors in each container). There are two possible ways to change the total number of actors by changing the number of containers by changing the number of actors in each container. Specifically, we test with the following configurations: 1) 3 containers with 13 actors inside each, 2) 3 containers with 8 actors inside each, 3) 3 containers with 2 actors inside each, 4) 2 containers with 13 actors inside each and 5) 1 container with 13 actors inside each.

Results are shown in Fig. 5. On `corridor`, the performance improvement provided by the increasing number of actors is significant when there 39 actors (CMARL). `CMARL_8_actors` and `CMARL_2_containers`, with totally 24 and 26 actors, can explore winning strategies while other ablations with fewer actors can not win.

On `academy_counterattack_hard`, perhaps interestingly, the performance of our method increases linearly with the total number of actors, regardless of the configuration. CMARLwith 39 actors can

win 30% of the games. `CMARL_8_actors` and `CMARL_2_containers`, with totally 24 and 26 actors, achieve a win rate of around 20% after 1300 minutes of training.

For (2), to test whether exploring with different policies is efficient, we carry out experiments by running our method with all actors using centralised learner's policy, which is periodically copied from the centralised learner, to collect new experience. Results are shown in Fig. 5. We can see that, on both `corridor` and `academy_counterattack_hard`, the ablation underperforms CMARL, consolidating the function of the diversity-encouragement learning objective.

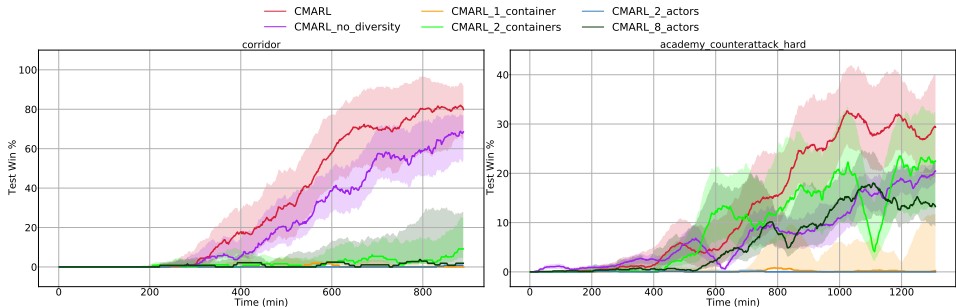

Figure 5: Comparisons of ablations against CMARL.

## 4.5 COMPARISON WITH OTHER DISTRIBUTED METHODS

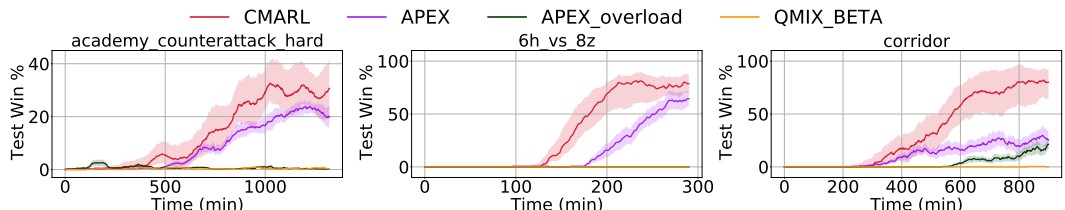

Figure 6: Comparisons of other distributed methods against CMARL.

We compare our method against other distributed methods which use multiple actors to collect experience. Since there is not much previous research in distributed MARL, for multi-agent distributed baselines, parallel QMIX (Rashid et al., 2018) with 39 actors collecting experience is tested on the same devices as our method. We call this method QMIX-BETA. We also compare our methods with single-agent distributed RL algorithms, APE-X (Horgan et al., 2018), by directly applying it to MARL settings. Since the computation cost is larger in MARL, when running APE-X with more than 14 actors on `corridor`, 16 actors on `6h_vs_8z`, and 30 actors on GRF, the CPU will be overloaded. We call APE-X running with this amount of actors APEX-overload. In the meantime, we also run APE-X with 10 actors, where CPUs operate normally (does not overload). We call this baseline APEX.

Because QMIX-BETA's learner and actors cannot uninterruptedly learn or collect experience, it cannot learn a winning strategy in the given training time. APEX-overload has a very high CPU usage, so the learning and experience collecting is slow. As a result, its performance is worse than APEX. APEX has a lower performance than CMARL on all environments, especially on `corridor`, where the size of a trajectory is the biggest among three maps. This suggests that our method can well deal with large scale tasks.

## 5 CONCLUSION

In this paper, we propose a distributed learning framework for value-based multi-agent reinforcement learning. We use a containerized architecture consists of several containers and a centralized learner. Each container interacts with several environment instances and use a specially designed multi-queue

manager to avoid blocking. Experience with high priority are sent to the centralized learning, so that data transfer overhead is significantly reduced. Moreover, trajectories collected from each container are encourage to be as diverse as possible. With these novelties, our method achieves impressive performance on both GRF and SMAC benchmark.

**Reproducibility**    The source code for all the experiments along with a README file with instructions on how to run these experiments is attached in the supplementary material.

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

## A  COMPARE WITH PARALLELIZED PYMARLV2

We test parallel QMIX with 39 actors, run on the same device as ours, and adopt all code level modifications of PyMARL-v2. We call this version QMIX-BETA-v2. We run an experiment on `academy_counterattack_hard` to compare our method against QMIX-BETA-v2. The result is shown in Fig. 7. We observe that after 1250 minutes of training, QMIX-BETA-v2 achieves a win rate of about 10%, while ours can achieve 30%.

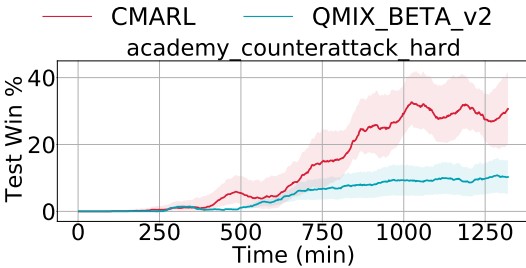

Figure 7: Comparisons of parallel PyMARL-v2 implementation of QMIX against CMARL.

## B  EXPEND BASELINE RUNNING TIME FOR SMAC

Baselines like RODE and CDS can solve super hard maps of SMAC. We extend the training time of these two baselines on four super hard maps on SMAC. The results is shown in Fig. 8.

RODE is 1.25 times slower than our method on `3s5z_vs_3s6z` and 2.67 time slower on `MMM2`. It has a trend of learning to win, but the performance is still far below our method on `6h_vs_8z` and `corridor` even with time expended. CDS also underperforms our method, being 1.5 times slower than our method on `corridor` and far below our method on `3s5z_vs_3s6z`, `6h_vs_8z` and `MMM2`.

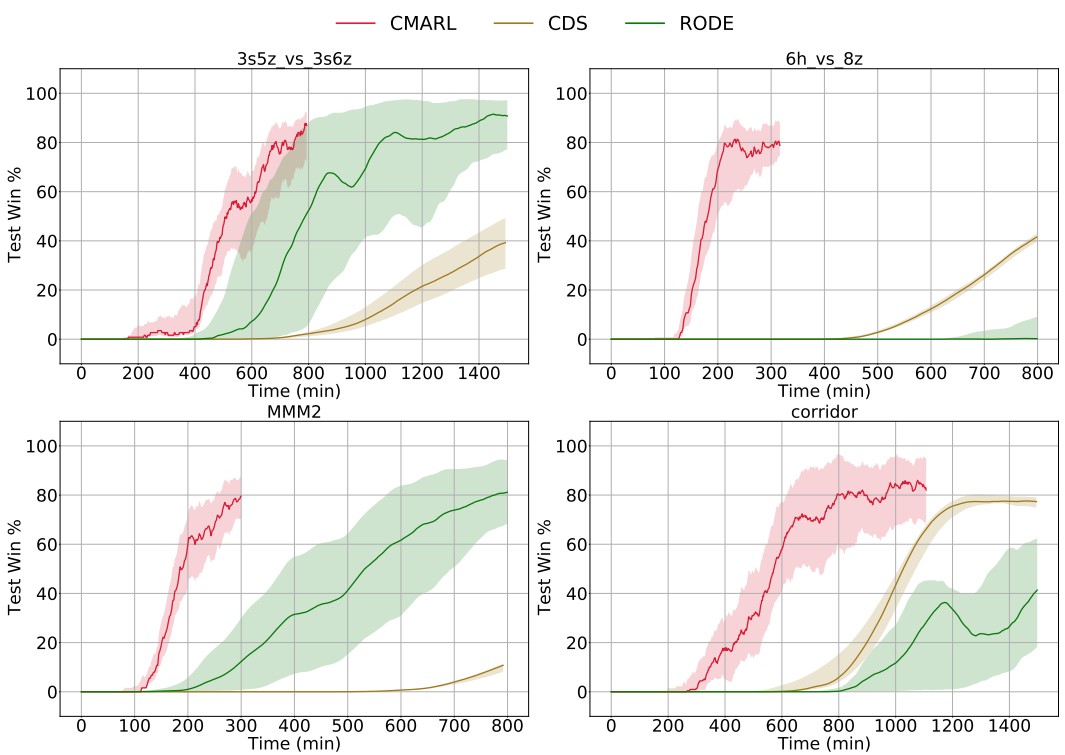

Figure 8: Expend running time of baselines.

