# OpenReview forum: "Containerized Distributed Value-Based Multi-Agent Reinforcement Learning"
_ICLR.cc/2022/Conference — ICLR 2022 Submitted_

### Official Review · Reviewer_Qy4m · 2021-11-01

**Correctness:** 3
**Technical Novelty And Significance:** 3
**Empirical Novelty And Significance:** 2
**Recommendation:** 3
**Confidence:** 3

**Main Review:**


 ~ Positives
 - The paper is well written and conveys its ideas clearly. The method is described in a way that is clear, motivates most of the architectural decisions, and seems to be reproducible.
 - The authors provide code with their submission which makes the experiments reproducible. The code is built on top of the PyMARL repository which is well-known and easy to use.
 - Encouraging diversity between containers, to my knowledge, is novel. It also seems to offer a significant advantage when compared to the no-diversity baseline in the experiments. I'd be interested in further experiments regarding diversity, e.g.:
 	- is there a number of containers after which diversity doesn't matter? Experiments seem to use two or three containers. If you were to use a much larger number, would that affect the usefulness of the diversity objective?
 	- Some layers are shared across the containers and the centraliser. If they weren't shared, would they more easily develop diverse policies? Since containers use local learners I expect some diversity even without the specific loss.


 ~ Negatives

- My main issue, and the reason I cannot at this stage recommend acceptance of this work, is the presentation of the experiments section. The presented experiments use time (in minutes) in the x'x axis - and this leads to CMARL looking significantly better than the baseline methods. Specifically, in figure 4, many of the baselines do not appear to learn at all and appear to converge to significantly lower returns than CMARL. However, this is not the case: Papoudakis et al[1] show QMIX (and other methods) to completely solve (close to 100% win-rate) tasks such as 3s5z and MMM2. I understand that it might be the case that those algorithms are significantly slower in wall-clock time, and might just not have enough time to complete their training, but it should still be clear from the figures that they do, in fact, converge to much higher values than shown.

This leads me to the second issue of the experiments section: the selected baselines were not supposed to be fast, but rather sample efficient. In the related works section, the authors discuss distributed algorithms that focus on speed. I believe comparisons with these methods would be fair and offer a better comparison with existing literature. For instance, IMPALA implementations in MARL exist and could be applied directly to SMAC and possibly GRF.

 That said, my knowledge of the GRF environment is limited, but the results do seem exciting.

- I am unsure how the presented method makes use of the multi-agent nature of the problem. Yes, the underlying algorithm is multi-agent (QMIX), but my understanding is that the containerized solution is agnostic to the algorithm. Is it also possible to apply this as-is in single-agent settings or have I misunderstood this?
If the multi-agent setting is only used as a motivation to the increased observation sizes/data requirements, then the experiments should justify this. But now, the observation sizes of these environments should be smaller compared to image-based single-agent environments.

 ~ Other

 - page3/first line: It is not typical for MARL to also include the global state. It is indeed very common in the centralised training/decentralised execution paradigm, but there are plenty of multi-agent settings where global state is never observed. I believe the phrasing here should change.

 - Sharing (some) parameters between the global and local learners is an interesting idea and is motivated in the paper. I would be interested in seeing how parameter sharing across agents is handled here. In QMIX parameters are always shared across agents [2], is this also the case for CMARL? The authors could discuss how containers can handle parameter sharing across agents.


 [1] Georgios Papoudakis, Filippos Christianos, Lukas Schäfer, & Stefano V. Albrecht. Benchmarking Multi-Agent Deep Reinforcement Learning Algorithms in Cooperative Tasks, Proceedings of the Neural Information Processing Systems Track on Datasets and Benchmarks (NeurIPS), 2021

 [2]Tabish Rashid, Mikayel Samvelyan, Christian Schroeder Witt, Gregory Farquhar, Jakob Foerster, and Shimon Whiteson. Qmix: Monotonic value function factorisation for deep multi-agent reinforcement learning. In International Conference on Machine Learning 2018.


**Summary Of The Paper:**

The authors propose a framework for distributed multi-agent reinforcement learning. By grouping actors, replay buffers, learners, and a queue manager into containers CMARL can be used to scale learning to the requirements of the multi-agent setting.

**Summary Of The Review:**

The paper presents clearly some interesting ideas that improve the training time of MARL algorithms. However, the presentation of their results is lacking in both baselines and in metrics (convergence or sample efficiency). At this stage, I cannot recommend this paper for acceptance.

---

> ### Author Response · Authors · 2021-11-21
> **We discuss how we make use of the multi-agent nature of the problem and how parameter is shared across agents. Empirically, we present the requested experimental results.**
>
> Thanks for your valuable review.
>
> > **"The presented experiments use time (in minutes) in the x'x axis - and this leads to CMARL looking significantly better than the baseline methods."**
>
> We agree that sample efficiency is important for MARL algorithms, but this paper focuses on another important metric, i.e., training speed. Although distributed training has been widely studied in the single-agent RL setting, it remains largely unstudied in the multi-agent setting. To our best knowledge, this paper is one of the first works to study distributed training for value-based MARL. We sincerely hope the reviewer can recognize the significance of training time in practice and re-evaluate this work.
>
> > **"Specifically, in figure 4, many of the baselines do not appear to learn at all and appear to converge to significantly lower returns than CMARL."**
>
> To further clarify the reviewer's concern, we expend running time of baselines and show the experiment results in Fig.8. (Appendix B). The results show that: RODE is $1.25$ times slower than our method on $\mathtt{3s5z\\_vs\\_3s6z}$ and $2.67$ time slower on $\mathtt{MMM2}$. It has a trend of learning to win, but the performance is still far below our method on $\mathtt{6h\\_vs\\_8z}$ and $\mathtt{corridor}$ even with time expended. CDS also underperforms our method, being $1.5$ times slower than our method on $\mathtt{corridor}$ and far below our method on $\mathtt{3s5z\\_vs\\_3s6z}$, $\mathtt{6h\\_vs\\_8z}$ and $\mathtt{MMM2}$.
>
> > **Compare with distributed reinforcement learning methods.**
>
> We compare with other distributed methods, including parallel value-based MARL methods like QMIX-beta and distributed single-agent methods like APE-X. Results are shown in Fig. 6, section 4.5. We can see that our method significantly outperforms these baselines.
>
> > **"I am unsure how the presented method makes use of the multi-agent nature of the problem."**
>
> Our work aims to enable efficient distributed training for collaborative multi-agent reinforcement learning (MARL) to significantly improve its training speed. As discussed in this paper, although distributed training has been extensively studied in single-agent RL, multi-agent RL poses new challenges, including demanding data transfer, inter-process communication, and exploration. Our containerized method effectively addresses these challenges.
>
> Moreover, the data transfer is largely increased in multi-agent settings. In Atari, the commonly used training paradigm is to use a $4\times 84\times 84$ image as state. DQN uses minibatches of size $32$ to train. Each element inside the batch consists of the current state and the next state, which is of size $8\times 84\times 84=56448$ in total. (There may also be overlap frames between current and next state, so the value could be even smaller.) So a batch is of size no more than $32\times 8\times 84\times 84=1806336$. In MARL, QMIX uses batches consisting of 32 trajectories to train. Take $\mathtt{corridor}$ as an example, one trajectory consists of $400$ maximum timesteps. And at each time step, there are $6$ agents' observation, each of size $156$, and $1$ global state of size $282$. So the batch is of size $32\times400\times(6\times156+282)=15590400$. Thus, for one single element inside the batch, the size in MARL environments is about $8.6$ times bigger than image-based single-agent RL environments.
>
> > **"The authors could discuss how containers can handle parameter sharing across agents."**
>
> In our method, agents share parameters in the same way as QMIX, i.e. agents use the same parameter to compute individual Q value.

---

> > ### Comment · Reviewer_Qy4m · 2021-11-29
> > **Reply to authors**
> >
> > I agree that training time is important, but the comparison is still unfair and does not clearly convey the differences of the algorithms. For instance the CPU time between the algorithms is certainly very high. At this point, the figures are comparing an algorithm that uses dozens of processes in parallel with algorithms that use a single one - and it's not surprising that using parallel processes works much faster.
> >
> > The comparison with APE-X is a step in the right direction, and we can clearly see the performance gap has decreased significantly. I believe an extended comparison with such methods (in all the environments) makes much more sense and should replace the existing main experiment section.
> >
> > "I am unsure how the presented method makes use of the multi-agent nature of the problem." -> The authors do not answer my question. I agree that MARL often needs more compute (although the comparison offered here is unfair because passing the whole episode as done here, is to bypass the partial observability which is not MARL specific - I can easily come up with a single-agent environment that needs more data that corridor). But still, my question was targeting the approach of the problem and not the motivation itself.
> >
> > I do not think the paper has improved significantly and the answers were not satisfying - I will keep my previous score.

---

### Official Review · Reviewer_uCBF · 2021-11-01

**Correctness:** 3
**Technical Novelty And Significance:** 2
**Empirical Novelty And Significance:** 2
**Recommendation:** 5
**Confidence:** 3

**Main Review:**

Strengths:
The presentation of the paper is mostly clear, with each section describing exactly what the corresponding component in CMARL does. Graphical illustrations are also helpful in showcasing the neural network structures.
The experiment setup is detailed enough for reproduction, and the algorithm has been tested across different environments with respect to the current state-of-the-art algorithmic models. Ablation studies have been carried out on different components of CMARL.


Weaknesses:
The degree of novelty seems relatively low. On a high level, CMARL resembles a hierarchical stack of two QMIX-like MARL algorithms, with container-level outputs of Q values serving as the input of higher-level centralizer. The multi-queue manager doesn’t seem to be too different from the standard prioritized experience replay mechanism, and more explanations on the novelty of the paper is needed.
It is unclear how the container achieves ‘diversity’ in Section 2.3. The additional mutual information loss doesn’t provide much intuition on how the experiences will vary.
While the experiments cover a broad range of environments, it would still be ideal to incorporate additional graphs(e.g. Goal difference vs epoch in addition to goal difference/win rate vs time) to complement existing results. Moreover, the ablation studies seem inadequate - why the specific choices of parameters? How do the ablation scenarios in Table 1 compare to each other, when the total number of agents seem to vary in different cases? More explanations should be provided.


**Summary Of The Paper:**

This paper proposes a distributed containerized multi-agent reinforcement learning(CMARL) framework that addresses three challenges in MARL: intra-agent observation data transfer, inter-process communication,and efficient coordinated exploration amongst the agents.  Using a container that collects environment experiences from parallel actors into buffers and learns local policies, CMARL demonstrates notable performance improvements with respect to time as compared to state-of-the-art benchmarks.


**Summary Of The Review:**

The paper does lead to some improvement in the field of multi-agent reinforcement learning, although the techniques adopted seem relatively incremental. Furthermore, explanations on experimental parameter choice and novelty should be included.  I’d be happy to change my scores, if the authors address such concerns accordingly.

---

> ### Author Response · Authors · 2021-11-21
> **We discuss our novelty, and explain how we encourage diversity and why we choose the parameters.**
>
> Thanks for your valuable review.
>
> > **Novelty of our method.**
>
> If we directly apply single-agent distributed methods like APE-X to multi-agent tasks, running more than 14 environment instances on $\mathtt{corridor}$ and 30 environment instances on $\mathtt{academy\\_counterattack\\_hard}$ would overload 80 CPU cores and 1 GPU. Technical novelties are required to enable more parallelism. The core challenge is how to efficiently handle and learn from a much larger volume of experience. The _first novelty_ of our method is the containerized structure, which reduces inter-device data transfer by locally selecting promising experience with high priority. If all local learners share the same parameters, similar behavior among actors would be induced, reducing the probability that more talented behavior is found. Our _second novelty_ solves this problem by partially sharing parameters and encouraging container-level diverse joint behaviors. Additionally, compared to single-agent distributed methods, the containerized structure enables efficient use of GPUs because a container can reside on GPUs, which is also a contribution of our approach. Furthermore, since typically many actors reside in a single container, data transfer within a container is still demanding. Our _third novelty_ focuses on data transfer control in a container. The structure and details of the multi-queue manager and buffer manager are specially designed for this purpose.
>
> In summary, the contribution of this paper is that we pinpoint challenges in realizing distributed MARL and design the first value-based distributed MARL method that adopts a novel container structure with efficient inter- and intra-process data transfer control and a diversity-encouraging learning objective.
>
> > **”It is unclear how the container achieves ‘diversity’ in Section 2.3.”**
>
> When the mutual information between the joint trajectory and the variable of container ID is optimized, the joint trajectory would contain more information about the container ID, which means that joint trajectories are identifiable by contain IDs. Therefore, the joint behavior of agents are different in different containers, leading to diverse policies that promote exploration.
>
> > **”The ablation studies seem inadequate - why the specific choices of parameters?”**
>
> We carry out ablations to study how the container structure, the diversity loss, and the number of actors contribute to the final performance of CMARL. We explain why we choose the parameters in Table 1 in our ablations.
>
> In CAMRL, we use 3 containers each with 13 actors. To test the influence of the number of actors, we reduce the number of actors in two ways — (1) reducing the number of containers and (2) reducing the number of actors in each container.  For (1), we reduce the number of containers to 2 and 1. For (2) we reduce the number of actors to 8 and 2.

---

### Official Review · Reviewer_axQN · 2021-11-01

**Correctness:** 2
**Technical Novelty And Significance:** 2
**Empirical Novelty And Significance:** 2
**Recommendation:** 3
**Confidence:** 3

**Main Review:**

Edit: Thanks for providing the clarifications and new experimental results. However, I'm still not convinced of the new experimental results. According to fig.6 and fig.7, the result of QMIX_BETA_v2 without PER is much better than QMIX_BETA with PER, so I recommend all the experiments on both SMAC and GRF taking pymarl-v2 as the benchmark. I believe that the paper could be considerably improved. I keep my score.

**Strengths**:

- The motivation is clear. The paper clearly argues that a successful framework for distributing acceleration has been missing from prior work on multi-agent RL.
- The implementation of distributing has some technical novelties.
- The architecture consisting of container and centralizer can ease the demanding data transfer and accelerate the training process.
- With lots of training resources, this method can obtain better experimental results.
- This paper is also well organized and clearly written.

**Weaknesses**:

- In Subsection 2.3, the approximate calculation of $p(\mathbf{a}_t \mid \mathbf{\tau}_t)$ is wrong. If we approximate $p\left(\mathbf{a}_{t} \mid \mathbf{\tau}_{t}\right)=\frac{1}{N} \sum_{id=1}^{N} p\left(\mathbf{a}_{t} \mid \mathbf{\tau}_{t}, id\right)$, with the Boltzmann SoftMax distribution replacement of $p\left(\mathbf{a}_{t} \mid \mathbf{\tau}_{t}, id\right)$ and $\mathbf{\pi}_{i d}\left(\mathbf{a}_{t} \mid \mathbf{\tau}_{t}\right)=\Pi_{i=1}^{n} \pi_{id}\left(a_{t}^{i} \mid \tau_{t}^{i}\right)$, the approximate equation should be $p(\mathbf{a}_t | \mathbf{\tau}_t) \approx \frac{1}{N} \sum_{j=1}^{N}\left(\Pi_{i=1}^{n} \pi_{j}\left(a_{t}^{i} \mid \tau_{t}^{i}\right)\right)$, not $\Pi_{i=1}^{n}\left(\frac{1}{N} \sum_{j=1}^{N} \pi_{j}\left(a_{t}^{i} \mid \tau_{t}^{i}\right)\right)$. Moreover, there is a minor mistake, that Eqn.(6) uses the symbol $\pi_{id}\left(a_{t}^{i} \mid \tau_{t}^{i}\right)$, while the parameter explanation of Eqn. (5) uses $\pi_{i d}^i\left(a_{t}^{i} \mid \tau_{t}^{i}\right)$, which should be the same.

- The implementation encouraging diversity among containers is similar to CDS[2]. Eqn. (3) is exactly the same, with the different meaning of "id". CDS aims to encourage diversity among agents, while CMARL aims to diversify containers. Meanwhile, based on the analysis in (1), Eqn. (6) and Eqn. (7) need to be corrected.
- The design of a multi-queue manager is equivalent to a two-level replay buffer, which I think is an implemented trick. When the learner samples data, the prepacked batch data are sent to it, making the two processes, leaner sampling data and actor inserting data, complementary interference. However, when prepacking in a multi-queue manager, it also faces the problem of the mutual exclusion of reading and writing to the same storage area, which is inevitable. Therefore, the role of the multi-queue manager is mainly related to the frequency of inserting and sampling experience, i.e., the number of actors, network size, and so on. On the other side, it also brings a new problem: when the learner performs data sampling, the newly generated data cannot be collected in the first time.
- Only trajectories with high priority are sent to the centralizer learner. The computation of the priority is $p_{\tau}=\operatorname{Normalize}\left(\sum_{t} r_{t}\right)+\epsilon$, where $\operatorname{Normalize}\left(X\right) = \frac{X-L}{H-L}$. However, It's unclear that why to use this computation to judge the quality of experience. And I also wonder about the result of QMIX using the same method to screen experience, with multiple actors generating trajectories.
- Finally, I think it's unfair to compare CMARL and other baseline algorithms with the same training time. CMARL uses more resources to train a better model. For example, QMIX only has 1 actor, but CMARL has 39 actors, which means during the same time, the data generated by CMARL is 39 times higher than QMIX. Besides, because of the diversity of different containers, the centralizer and containers have different parameters, so the number of parameters is also higher than QMIX. I only take QMIX as an example, and other baseline algorithms also face the same problems. The paper RIIT[1] recently shows that QMIX could achieve SOTA performance among all SMAC maps with some code level modification only, including the number of actors. So, I recommend the authors reimplement based on pymarl-v2 (I call the optimized version of pymarl by Jian Hu in paper RIIT[1] as pymarl-v2) and show how it is used performs.


[1] Hu J, Jiang S, Harding S A, et al. RIIT: Rethinking the Importance of Implementation Tricks in Multi-Agent Reinforcement Learning[J]. arXiv preprint arXiv:2102.03479, 2021.

[2] Chenghao Li, Chengjie Wu, Tonghan Wang, Jun Yang, Qianchuan Zhao, and Chongjie Zhang. Celebrating diversity in shared multi-agent reinforcement learning. arXiv preprint arXiv:2106.02195, 2021.

**Summary Of The Paper:**

This paper introduces a new distributed value-based multi-agent reinforcement learning framework to solve problems faced by multi-agent tasks. It divides the system into two parts, multiple containers, and one centralizer. Containers are trained with trajectories generated by their own actors interacting with the environment. In contrast, the centralizer is trained with high-priority samples sent by all containers, easing the demanding data transfer. Besides, this paper designs a multi-queue manager between actors and replay buffer to avoid inter-process communication blocking. Furthermore, this paper proposes a new loss function encouraging different containers to be diversified to promote exploration. Finally, it achieves better results on both Google Research Football and StarCraft II micromanagement benchmark.

**Summary Of The Review:**

I think this paper introduces a new distributed value-based multi-agent reinforcement learning framework to ease the demanding data transfer and accelerate the training process, which is a benefit of large-scale training. However, this paper has some fatal flaws:
- The approximate equation in subsection 2.3 is wrong.
- The diversity design lacks innovation with a fatal error.
- The design of a multi-queue manager needs to weigh the pros and cons.
- The basis for computing priority is unclear, and another ablation study is necessary.
- The experiment needs a more convincing comparison.

Some other comments: Figure 1 need more clarification and refinement. Too many colors are used and the pipeline is unclear. Its title also has no explanations.

---

> ### Author Response · Authors · 2021-11-21
> **We explain our approximation in deversity design and the pros/cons of the multi-queue manager. Empirically, we present the requested experimental results.**
>
> Thanks for your valuable review.
>
> > **"The approximate equation in subsection 2.3 is wrong."**
>
> We thank the review to point out this mistake. We actually approximate $p(a_t|\\tau_t)=\Pi_{i=1}^n p^i(a_t^i|\\tau_t^i)\approx\Pi_{i=1}^n(\frac{1}{N}\sum_{j=1}^N\pi_j(a_t^i|\\tau_t^i))$. We update it in our revision.
>
> > **"The implementation encouraging diversity among containers is similar to CDS."**
>
> Though sharing some similarity, our approach and CDS optimize different objectives. We focus on diversity of collective behavior among different containers, and CDS optimize diversity among the behavior of individual agents.
>
>
> > **"The design of a multi-queue manager is equivalent to a two-level replay buffer, which I think is an implemented trick."**
>
> The multi-queue manager and the buffer manager are not just an implemented trick because they are specially designed to achieve the following goals.
>
> (1) Enabling uninterrupted learning of the learner. These two managers significantly reduce the probability of the learner waiting between two model updates by ensuring the learner can sample batches from the queue almost with no waiting when it is going to do a new model update. The communication queue which sends batches to the learner always contains at least a certain amount of batches when the buffer manager is sampling. So if the learner does not use all these batches to learn during the time when the buffer manager inserts new experiences, the learning is not interrupted.
>
> (2) Enabling actors to spend more time collecting new experiences. Actors' waiting time, which is caused by too many new experiences in the communication queue that actors use to transmit new experiences to the buffer manager, is significantly reduced. The multi-queue manager spends more time on receiving new experiences than the buffer manager, which makes actors' communication queue that transmit new experiences to the multi-queue manager is less likely to be full than transmit to the buffer manager.
>
> (3) The mutual exclusion of reading and writing problem can be avoided because, when the multi-queue manager is prepacking all new experiences, it does not receive new experiences temporally. The multi-queue manager receives new experiences and prepack them alternatively.
>
> We also run an experiment on $\mathtt{6h\\_vs\\_8z}$ to verify (1) and (3). In the experiment, our method's learner spends about $90\\%$ time learning and actors spend about $99\\%$ time collecting experiences. If all actors directly send new experiences to the buffer and the learner directly samples batches of experiences from the buffer, the learner only spends about $50\\%$ time learning and actors spend about $90\\%$ time collecting experiences.
>
> > **"It also brings a new problem: when the learner performs data sampling, the newly generated data cannot be collected in the first time."**
>
> It is true that the newest data generated by actors is not immediately used by the learner to train. However, this is not a problem, because this data is still used by the learner once it is inserted to the buffer, and the ''life'' time that it is inside the buffer is not affected by our multi-queue manager or buffer manager. And also, even without our managers, the newly generated experience still needs some time to be used for learning, because it needs to be transferred and inserted in the buffer first, and then it needs to be sampled in a batch, and finally it also needs to be transferred to the learner.
>
> > **"I also wonder about the result of QMIX using the same method (of computing priority) to screen experience, with multiple actors generating trajectories."**
>
> We carry out the experiment as suggested by the reviewer. The result is updated in section 4.5 of the revised version of the paper.The parallel version of QMIX (QMIX-beta) uses the same method of priority calculation and the same amount of resources. Empirical results show that CMARL significantly outperforms this baseline.
>
> > **"QMIX only has 1 actor, but CMARL has 39 actors. ... I recommend the authors reimplement based on pymarl-v2."**
>
> We parallel QMIX with 39 actors, run on the same device as ours, and adopt all code level modification of pymarl-v2. We call this version QMIX\_BETA\_v2. We run an experiment on $\mathtt{academy\\_counterattack\\_hard}$ to compare our method against QMIX\_BETA\_v2. The result is shown in Fig.7. (Appendix A). We observe that after 1250 minutes of training, QMIX\_BETA\_v2 achieves a win rate of about $10\\%$, while ours can achieve $30\\%$.

---

### Official Review · Reviewer_HX1e · 2021-11-02

**Correctness:** 2
**Technical Novelty And Significance:** 2
**Empirical Novelty And Significance:** 2
**Recommendation:** 5
**Confidence:** 4

**Main Review:**

Although this paper focuses on a very interesting issue, there are still some places in this paper that make me confuse. At present, I think the author needs to further clarify my confuse.

Strengths:
1) The field of the paper is very important and interesting
2) The writing of the paper is clear, and the method is intuitive and effective

Weaknesses:
1) The author raised the problem of demanding data transfer at the beginning, especially when the GPU and CPU copy each other. I was very excited about this at the beginning, because this problem is very important. This problem exists in all (MA) RL framework based on cpu simulation, unless the environment is run on the GPU or the transmission effect between the CPU and the GPU is improved from the hardware. Otherwise, there is no particularly good method intuitively. But then I was very disappointed. The author did not solve this problem. The author just placed the environmental interaction part in the container. In fact, once the GPU is used for deep learning training, both the container and the centralizer need to copy the data of the cpu and gpu.
2) For the Inter-process communication problem, the author proposed multi-queue manager and buffer manager to optimize, so as to meet the author's proposal: "A critical design consideration of a successful distributed reinforcement learning framework is the uninterrupted learning of learners. ". However, the author lacks a detailed discussion of this module. Judging from the current version, the author only used "signal" to control read and write conflicts. This seems intuitive, but it does not satisfy the "uninterrupted learning of learners".
3) For the problem of effective exploration, the author adopts an MI-based method to encourage the container to have different policies for efficient exploration. Some methods under this part of single agent RL have already had similar work, such as DIYAN [1], etc., and the use of similar population exploration strategies has also been studied in single agent rl, such as Fig 2 in [2].
4) The author uses KL of different policies to constrain different policies to have different (exploratory) behaviors. From the current version, this part of the policies is in different containers. Intuitively, this will increase the load after parallelization. However, the author does not discuss.
5) Although the author emphasizes that their framework is distributed MARL, the author's distributed approach is to treat the marl problem as a joint single agent, which means that different single agnet distributed training schemes are also a very important comparison  method, but the author did not compare.
6) The author claims to have solved google football. After I checked the author's code, I have a doubt to be clarified: the author uses "scoring" to calculate the reward, which means that the positive reward will be very sparse, so that PER can be efficiently filter out "useful" trajectories, but the comparison method does not have a similar mechanism, which seems very unfair.
7) It is also the experimental part. The current version seems to be that the author’s method uses more resources for training. For better comparison, the control experiment should also have corresponding computing resources, such as using more parallel environments. .

[1] DIVERSITY IS ALL YOU NEED: LEARNING SKILLS WITHOUT A REWARD FUNCTION

[2] Interactive Parallel Exploration for Reinforcement Learning in Continuous Action Spaces


**Summary Of The Paper:**

This paper focuses on an interesting and important question: how to perform distributed multi-agent deep reinforcement learning? The author first proposed three challenges to be considered: 1) Demanding data transfer. 2) Inter-process communication. 3) Effective Exploration. Further, the author proposed a container-based distributed marl framework, by placing the part that interacts with the environment in the container relieves the pressure of the cpu, and at the same time encourages different containers to have different policies and uses PER to select high-value samples, thereby improving training efficiency.

**Summary Of The Review:**

Although this paper focuses on a very interesting issue, there are still some places in this paper that make me confuse. At present, I think the author needs to further clarify my confuse.

---

> ### Author Response · Authors · 2021-11-21
> **We discuss data transfer management and explain why our learner can uninterruptedly learn. Empirically, we compare CMARL with other distributed methods. We also address other concerns.**
>
> Thanks for your valuable review.
>
> > **"The author did not solve this problem. The author just placed the environmental interaction part in the container. In fact, once the GPU is used for deep learning training, both the container and the centralizer need to copy the data of the cpu and gpu."**
>
> It is inevitable that data need to be transferred between devices if environments and actors run on different devices. However, this amount of transfer is relatively small when comparing to data transfer between different processes. Data transfer between different processes includes sending new experiences to and receiving batches from both container's and centralizer's buffer. Comparing to this, the total memory size of CPU/GPU transfer in all actors is only $3$\% the memory size of inter-process data transfer.
>
> > **"The author only used "signal" to control read and write conflicts. This seems intuitive, but it does not satisfy the uninterrupted learning of learners."**
>
> To not interrupt learner's learning process, one needs to make sure the learner performs the next model update as soon as possible when the learner finishes the current model update. Performing a model update requires sampling a batch of experience. However, when a model update is finished, sampling directly from the buffer may not be allowed because the buffer may update itself with new experiences at that time. To solve this problem, we introduce the buffer manager which samples batches of experiences and send them to the learner through a communication queue in a way that the learner can always sample batches from the queue when it is going to do a new model update. To achieve this goal, we need to guarantee two conditions: (1) there are always experiences in the communication queue between the buffer manager and the learner and (2) when the learner reads from the queue, the requested memory is not write-protected. We achieve condition (1) by letting the buffer manager make sure that the communication queue contains at least a certain amount of batches when it is sampling. So if the learner does not use all these batches to learn during the time when the buffer manager inserts new experiences, condition (1) is achieved. In the meantime, condition (2) is guaranteed because we use a communication queue to send batches from the buffer manager to the learner, and this queue has a property that each batch is stored in a different place in memory. This property ensures that there is no read and write conflict between reading a batch from one place and inserting another batch on another place.
>
> We also run an experiment on $\mathtt{6h\\_vs\\_8z}$ to verify our method. We observe that our learner spends about $90\\%$ of the time on learning. If all actors directly send new experiences to the buffer and the learner directly sample batches of experiences from the buffer, the learner only spends about $50\\%$ time on learning.
>
> > **"The author uses KL of different policies to constrain different policies to have different (exploratory) behaviors. Intuitively, this will increase the load after parallelization.**"
>
> As we derived in equation (8), to encourage diversity during exploration, we only need to compute the KL divergence between agent $i$'s policy on $\tau_t^i$ from container $id$ and that from container $j$. To compute such a term, we only need to compute the individual Q-values of agent $i$ from container $id$ and from container $j$ on $\tau_t^i$. We use a three layer network to compute the individual Q-value. The first two layers are shared between containers and the centralized learner, so we only need to maintain a global copy of every container's last fully-connected layer. Since the last fully connected layer has only a few parameters, this computation cost is relatively small.
>
> > **"The different single-agent distributed training schemes are also a very important comparison method, but the author did not compare."**
>
> We compare our method against APE-X and show the result in section 4.5 of the revised version of our paper. We observe that our method outperforms APE-X on both SMAC and Google Research Football tasks.
>
> > **PER for baselines.**
>
> We thank the reviewer for carefully reviewing our code. Using "scoring" to calculate the reward is a common practice [1, 2, 3]. Moreover, all experiments in section 4.5 use the same PER as our approach for a fair comparison.
>
> > **Baselines using the same amount of resources as CMARL.**
>
> In the revised version of our paper, we run parallel QMIX and APE-X using the same amount of resources as our method. Results are shown in section 4.5. We can see that CMARL still outperforms under this setting.
>
> [1] Kurach et al., Google Research Football: A Novel Reinforcement Learning Environment. AAAI, 2020
>
> [2] Mao et al., Neighborhood cognition consistent multi-agent reinforcement learning. AAAI, 2020
>
> [3] Li et al., Celebrating Diversity in Shared Multi-Agent Reinforcement Learning. NeurIPS, 2021

---

> > ### Comment · Reviewer_HX1e · 2021-11-30
> > **I keep my score**
> >
> > Thanks for providing the clarifications. Based on the overall discussion, I believe that the paper could be considerably improved, so I will keep my score as it is.

---

### Decision · Program_Chairs · 2022-01-20

**Decision:**

Reject

**Comment:**

This paper proposes a distributed containerized multi-agent reinforcement learning(CMARL) framework that addresses three challenges in MARL: 1) Demanding data transfer. 2) Inter-process communication. 3) Effective Exploration. Using a container that collects environment experiences from parallel actors into buffers and learns local policies, CMARL demonstrates notable performance improvements with respect to time as compared to state-of-the-art benchmarks.

Although the reviewers acknowledge that the paper addresses a relevant topic, proposes an effective method, and is well written, after reading the authors' feedback and discussing their concerns, the reviewers reached a consensus about rejecting this paper in its current form. They feel that the contribution is too incremental and that the experimental comparisons are somehow unfair.

I suggest the authors take into consideration the reviewers' suggestions while preparing an updated version of their paper for one of the forthcoming machine learning conferences.